# Encephalitis following COVID-19 Vaccination: A Systematic Review

**DOI:** 10.3390/vaccines11030576

**Published:** 2023-03-02

**Authors:** Mariam Abdelhady, Muhammad Ashraf Husain, Yousef Hawas, Mahmoud Abdelsalam Elazb, Lena Said Mansour, Mohamed Mohamed, Maya Magdy Abdelwahab, Ahmed Aljabali, Ahmed Negida

**Affiliations:** 1Faculty of Medicine, October 6 University, Giza 12585, Egypt; 2Medical Research Group of Egypt (MRGE), Cairo 11511, Egypt; 3Faculty of Medicine, Al-Azhar University, Cairo 11884, Egypt; 4Faculty of Medicine, Tanta University, Tanta 31511, Egypt; 5Faculty of Pharmacy, Al-Azhar University, Cairo 11884, Egypt; 6Internal Medicine Department, Damanhour Teaching Hospital, Damanhour 22511, Egypt; 7Faculty of Medicine, Zagazig University, Zagazig 44519, Egypt; 8Faculty of Medicine, Helwan University, Cairo 11795, Egypt; 9Faculty of Medicine, Jordan University of Science and Technology, Irbid 22110, Jordan; 10Department of Global Health, Harvard Medical School, Boston, MA 02115, USA

**Keywords:** COVID-19 vaccine, adverse effects, encephalitis, systematic review

## Abstract

Background: Since the advent of global COVID-19 vaccination, several studies reported cases of encephalitis with its various subtypes following COVID-19 vaccinations. In this regard, we conducted a systematic review to investigate and characterize the clinical settings of these reported cases to aid in physician awareness and proper care provision. Methods: We systematically searched PubMed, Web of Science, and Scopus and manually searched Google Scholar. Studies published until October 2022 were included. Demographic data, clinical features, vaccine data, treatment lines, and outcomes were extracted. Results: A total of 65 patients from 52 studies were included. The mean age of patients was 46.82 ± 19.25 years, 36 cases (55.4%) were males. AstraZeneca was the most-reported vaccine associated with encephalitis (38.5%) followed by Pfizer (33.8%), Moderna (16.9%), and others. Moat encephalitis cases occurred after the first dose of vaccination in 41/65 (66.1%). The mean time between vaccination and symptom onset was 9.97 ± 7.16 days. Corticosteroids (86.2 %) and immunosuppressants (81.5 %) were the most used lines of treatment. The majority of affected individuals experienced a full recovery. Conclusion: Our study summarizes the current evidence of reported post-vaccination encephalitis, regarding clinical presentation, symptoms onset, management, outcomes, and comorbid conditions; however, it fails to either acknowledge the incidence of occurrence or establish a causal relationship between various COVID-19 vaccines and encephalitis.

## 1. Introduction

Encephalitis is an inflammation of the brain tissues and is most usually caused by a viral infection (mainly herpes simplex virus), which represents about 75% of diagnosed cases; however, autoimmune causes such as N-methyl D-aspartate receptor (NMDAR) antibody encephalitis are also common [1]. Encephalitis is a neurological emergency that can cause severe cognitive impairment or death if not treated promptly. It can be diagnosed by at least two of the following criteria: fever, seizures, focal neurological findings by a cause of brain parenchymal damage, EEG findings indicative of encephalitis, lumbar puncture pleocytosis (more than four white cells per μL), or neuroimaging findings suggestive of encephalitis [2].

Recently, COVID-19 emerged as a new public health crisis affecting worldwide populations. As of the date of this review, 664 million confirmed cases and 6.7 million deaths have been reported since the outbreak in late 2019 [3]. To curtail the development of this disease, research on coronavirus diagnosis, prevention, treatment techniques, and vaccines was launched. The burden was heavily lifted when COVID-19 vaccines emerged. The mechanism of action of various vaccines aim to elicit immune response: the mRNA-based vaccines (PfiZerBioNTech and Moderna) are made up of genetically modified viruses RNA or DNA that produces a viral protein [4,5,6]. The genetically modified non-mRNA adenovirus vector vaccines (Janssen/Johnson and Johnson, Sputnik V, and AstraZeneca) also produce coronavirus proteins [5]. The spike protein or its fragments that resemble COVID-19 are introduced in protein subunit vaccines (Corbevax, Novavax). A killed or weakened COVID-19 virus is introduced in the attenuated viral vaccines Sinopharm and Sinovac Corona Vaccine [6].

Due to the urgency, vaccinations were approved based merely on the initial stages of clinical trials, without completion of all phases [7]. However, adverse reactions to vaccinations, including myelitis and severe disseminated encephalomyelitis, have been identified, although poorly documented [8].

Variable neurological complications after the COVID-19 vaccination, despite the unproven causes, have been reported. These include functional neurological disorder symptoms, such as altered mental status, autoimmune encephalitis (AE), acute disseminated encephalomyelitis (ADEM), dizziness, myalgia, fatigue, cognitive impairment, gait instability, facial palsy, Guillain–Barré syndrome (GBS), convulsions, strokes, transverse myelitis, chronic fatigue syndrome, and acute encephalopathy [9,10]. Recently, major neurological complications indicative of vaccination-related autoimmune encephalitis and acute encephalitis after the first dose of mRNA COVID-19 vaccines were reported [11,12,13,14,15]. Notably, acute disseminated encephalomyelitis (ADEM) was consistently reported after the viral vector-based vaccines or inactivated viral vaccine (AstraZeneca, Sputnik V, Sinopharm) [16,17,18,19,20,21,22].

Dutta et al. reported 19,529 neurological adverse events after COVID-19 vaccination, including encephalitis [23]. Zuhorn et al. demonstrated a temporal association between ChAdOx1 nCov-19 vaccination (AstraZeneca) and encephalitic symptoms [24]. The diagnosis of mRNA-1273 vaccine-induced encephalitis and status epilepticus was made by Fan et al. [25] in several recent cases.

The underlying mechanism of such symptomatology is not clearly understood; some researchers theorized that SARS-CoV-2 spike protein produced by mRNA-based vaccines may act as a catalyst for the inflammatory processes that ensue, particularly in autoimmune encephalitis [26].

Globally, vaccine hesitation is linked to a lack of trust in the COVID-19 vaccine’s safety and doubts about its effectiveness. However, vaccination acceptance rates increased to 75.2% last year according to an international survey [27]. Continuous vaccine improvement efforts and modifications are ongoing with the expanding range of immunity and adverse events in vaccinated populations. In this study, we aim to characterize clinical and laboratory features and the diagnostic and management implications of encephalitis cases following COVID-19 vaccinations to aid in physician awareness and proper care provision.

## 2. Methods

### 2.1. Database Search

Our systematic review followed the Preferred Reporting Items for Systematic Reviews and Meta-analyses (PRISMA) checklist. It is registered in PROSPERO database with ID number: CRD42023389901. We performed a systematic literature search of PubMed, Scopus, and Web of Science databases, from inception until October 2022, following the Preferred Reporting Items for Systematic Reviews and Meta-analyses (PRISMA) [28]. The following search strategy was used (COVID-19 vaccination OR SARS-CoV-2 vaccine OR COVID-19 vaccine) AND (encephalitis). To increase our chances of identifying all relevant studies, we manually retrieved other studies from Google Scholar and performed backward citation analysis.

### 2.2. Screening and Inclusion Criteria

We included all published or pre-published papers presenting cases of any type of encephalitis in individuals who received any type of COVID-19 vaccination either as case reports, case series, or letters to editors. No language restrictions were applied. Secondary studies including reviews and meta-analyses, book chapters, and press releases were excluded from our study. Studies underwent title and abstract blind screening by two reviewers using Rayyan Artificial Intelligence [29]. After the removal of duplicates, the identified full-text articles were examined, and we manually assessed retrieved full-text records from Google Scholar and related references of further studies (Figure 1).

### 2.3. Statistical Analysis

To provide a comprehensive understanding of the data included in those studies, we extracted patients’ characteristics (i.e., age and gender), the type of encephalitis, the type and dose of the vaccine, and the latency period before the onset of symptoms. We extracted symptoms, either relating to the nervous system or any other systems, and whether these patients had any other comorbidities. Investigations, treatment, and treatment outcomes were likewise extracted (Table 1). Extracted data were pooled into mean and standard deviation for continuous variables or frequency and percentage for categorical variables (Table 2).

### 2.4. Quality Assessment

For included case reports and case series, we used the Joanna Briggs Institute (JBI) quality assessment tools, based on the clinical features, history, diagnoses, interventions, and management plans. Fourteen letters to the editor were excluded from the assessment due to a lack of appropriate assessment tools. Two authors assessed the quality of included studies and resolved conflicts by consensus. For case reports, JBI domains included eight questions, and the case series checklist assessed ten domains. Grades were assigned such that; Low risk: 75–100%, Moderate: 50–74%, High risk < 50%. High-risk studies will be excluded.

## 3. Results

Out of 1395 studies identified from databases, 280 were excluded as duplicates. We screened 1079 records at a title and abstract level, through which we excluded 1030 records for irrelevancy. At this point, 15 studies retrieved from previous studies and 16 records identified manually through a Google Scholar search were compared with studies eligible for full-text screening (n = 49), excluding duplicates. In total, 80 records were assessed through full-text screening, from which 11 studies were excluded for reporting CNS infections other than encephalitis and non-COVID-19 vaccination; 14 systematic reviews, literature reviews, and meta-analyses were excluded; 3 studies were excluded for being high risk on quality assessment. In total, 52 studies were included in the final qualitative synthesis, see Figure 1.

### 3.1. Patient Characteristics

Patients’ mean age was (46.82 SD 19.25) years, 55.4% of patients were males (37/28), and one case presented as a transgender male. Twenty-four patients (36.9%) had several comorbidities, including hypertension, DM, PD, heart disease, hypothyroidism, polymyalgia rheumatica, polyallergy, herpes simplex, migraine, MS, irritable bowel, kidney disease, hyperlipidemia, SARS-CoV-2, Tolosa–Hunt syndrome, CLL, benign prostate hyperplasia, pulmonary embolism, mycoplasma pneumonia, vasculitis, and fibromyalgia. The mean time for symptoms appearance post-vaccination was 9.97 ± 7.16, which was mostly reported after the first dose (66.1%), followed by the second dose (29%), the booster dose (3.2%) and the third dose (1.6%), see Table 2.

### 3.2. Clinical Presentation

Of all 65 patients, 11 presented with Acute encephalitis (16.9%), 15 with Acute disseminated encephalomyelitis (ADEM) (23.1%), 4 with Acute hemorrhagic encephalitis (AHEM) (6.2 %), and 35 cases with other types of encephalitis (53.8%) encompassing: unspecified Autoimmune encephalitis, Anti-LGI1 encephalitis, Anti-NMDAR encephalitis, Meningoencephalitis, Acute encephalopathy, Herpes simplex encephalitis, Rasmussen encephalitis, Limbic encephalitis, Bickerstaff Brainstem Encephalitis, Brainstem encephalitis, Encephalomyelitis, Multifocal Necrotizing Encephalitis, Anti-GAD encephalitis, and MOG encephalomyelitis, see Table 1.

All 65 patients presented with both neurologic symptoms. Most occurring neurologic symptoms were fever in 23/65 (35.4%) and abnormal movements in 24/65 (36.9%), headache occurred in 20/65 (30.8%) patients, and seizure occurred in 15/65 (23.1%) patients, see Table 2. Other reported neurologic symptoms included dysarthria, aphasia, dysphasia, altered mental status, gait disturbance, cognitive decline, general weakness, hypophonia, ataxia, disturbed conscious level, paraplegia, numbness, areflexia, agitation, spasms, FBDS, behavioral disturbances, memory impairment, status epilepticus, Lhermitte’s phenomenon, paraparesis, hypoesthesia, sphincter dysfunction, Babinski sign, hallucinations, spontaneous defecation, psychosis, coma, urinary retention, diplopia, photophobia, psychological changes, hypoglossal nerve paralysis, dizziness, taste disorder, and facial nerve paralysis. Non-neurologic symptoms were present in 36 patients (55.3%) and included: ophthalmoparesis, ophthalmoplegia, papilledema, optic neuritis, photophobia, blurred vision, aspiration pneumonia, cough, palpitation, myocarditis, bradyphrenia, sinus tachycardia, cardiac pauses, silent myocardial infarction, atrial fibrillation, abdominal pain, constipation, diarrhea, dehydration, hypersomnia, myalgia, dizziness, back pain, fatigue, loss of appetite, ketoacidosis, sepsis, urinary tract infection, reactive arthritis, and skin rash. Three patients (4.6%) were hospitalized, see Table 1.

### 3.3. Investigations and Diagnostic Results

Diagnostic test results were primarily available for CSF and MRI findings; 61.5% (40/60) of MRI results were abnormal with the following findings: FLAIR, T2, and DWI hyperintensities in various regions, central focal hemorrhage, bilateral white matter lesions, minimal T2 sulcal hyperintensity without contrast enhancement, plaques in periventricular, juxtacortical and cortical areas, swelling and hyperintensities of the anterior part of the optic nerves, restricted diffusion through insular and mesial temporal cortices, swelling of the hippocampus, encephalomalacia in frontoparietal lobes, blurred gadolinium enhancement on T1-weighted images; for MRI spine: multiple enhanced lesions of the spinal cord were found in addition to longitudinal edema along the thoracic spinal cord with contrast enhancement and longitudinally extensive transverse myelitis. In CSF, the most common finding was pleocytosis (48.5 %). Seven patients (10.8%) had high protein in their CSF samples, and six had positive CSF antibodies, including ANA, Anti-LGI1, SARS-CoV-2 spike S1 RBD IgG, anti-NMDA, intrathecal IgA, and IgM.

### 3.4. Treatment Plan and Its Outcomes

All 65 patients received a spectrum of medical treatments (steroids, IVIG, antivirals, immunosuppressive drugs, plasmapheresis, antibiotics, anticonvulsants, and analgesics), and 2 patients received anticoagulants. Most patients, i.e., 56 (86.2%), received steroids and immunosuppressive drugs, while 53 (81.5%) received drugs such as rituximab and tocilizumab. In total, 15 received immunoglobulins (23.1%) and 10 received an antiviral treatment (15.4%). Only 9 patients received plasmapheresis (13.8%).

Overall, 41 patients made a full recovery (63.1%), 11 had residual symptoms (16.9%), and 9 were transferred to a rehabilitation facility for extensive residual symptoms, including hypophonia, disturbed conscious level, dysphagia, vegetative state or coma, short-term memory loss, and tonic-clonic seizures. Four patients (6.2%) died, see Table 2.

The type of vaccine administered was not statistically associated with any specific outcome (*p* = 0.124), see Table 3; however, patients with autoimmune encephalitis and other subtypes were more likely to undergo full recovery (*p* < 0.001), see Table 4. Patients who did not receive plasmapheresis were more likely to make a full recovery (*p* = 0.002), see Table 5 and Figure 2.

### 3.5. Quality Assessment

Fifty-two case reports and three case series were assessed using the JBI checklist, see Table 6. Most case reports (n = 40) had a low risk of bias, nine had a moderate risk, and three had a high risk of bias. All three case series had a moderate risk of bias; data are shown in Table 7. Three high-risk studies were excluded.

## 4. Discussion

In our sampled data, the majority of affected cases were males. According to the demographic distribution, encephalitis was likely to affect all ages, with the highest incidence in patients in their 40s. Most patients complained of encephalitis after the first dose 1–2 weeks post-vaccination. AstraZeneca was the most reported vaccine followed by Pfizer, Moderna, and others. Unlike typical diagnostic criteria for encephalitis (major criteria of patients presenting with altered mental status lasting ≥24 h with no alternative cause identified), the most occurring symptoms in our study were abnormal movements, fever, headache, and seizures [30]. CSF encephalitic findings and MRI abnormalities were evident in almost two-thirds of included patients. Most patients received corticosteroids as a part of the immunosuppressive regimen and achieved full recovery. However, death was common in patients who received plasmapheresis, which, to our understanding, could be contributed to the already late presentation and severe symptoms, which prompted plasma exchange therapy in the first place.

Although the exact etiology of vaccine-induced encephalitis is still not fully understood, it could be attributed to different potential mechanisms of vaccine-induced autoimmune diseases. COVID-19 vaccine was shown to trigger proinflammatory cytokine expression and the response of T-cells as in other vaccines [31]. This is important because these cytokines may reach the brain and activate microglial cells resulting in neuroinflammation [32]. However, the possible molecular mimicry between the vaccine antigens and self-antigens, or the acceleration of an ongoing autoimmune process caused by vaccines are still considered potential mechanisms [33].

Myalgia and general weakness were considered the most reported symptoms in a previous clinical trial assessing the safety of the AstraZeneca vaccine; notably, only two cases presented with neurologic symptoms, such as demyelinating polyradiculoneuropathy and hypoesthesia. Nonetheless, no encephalitis-associated symptoms were reported [34]. To our knowledge, no other clinical trials reported encephalitis post-vaccination. As the vaccinated population increased, several SARS-CoV-2 vaccines have been associated with neurological side effects on follow-up observational studies [35]. Most reported cases in the literature were immunized by AstraZeneca and Pfizer. The incidence of encephalitis after vaccination with the AstraZeneca and Pfizer-Biontech mRNA vaccines was estimated to be 8 per 10 million and 2 per 10 million vaccination doses, respectively, which denotes an extremely rare incidence of adverse event occurrence in comparison to other well-established vaccines such as hepatitis vaccines [24]. A previous systematic review addressing 11 patients with autoimmune encephalitis also corresponds with a greater majority of AstraZeneca and Pfizer predominant case reporting (8/15) [31]. Consistent with our findings, in which AstraZeneca and Pfizer comprised almost two-thirds of available cases. However, no statistical correlation between vaccine subtype and encephalitis outcomes could be established in our study. Generally, performing a lumbar puncture can suggest the presence of encephalitic involvement [36]. Huang et al. presented 9/11 cases with lumbar puncture findings indicating encephalitic changes. In our study, CSF sample abnormalities were indeed reported in most cases followed by MRI abnormalities. Full recovery is indicated in a greater percentage of all the aforementioned studies and also in our study.

### Strengths and Limitations

The main strength of this systematic review is the use of a thorough search strategy to locate studies for evaluation, minimizing selection bias. These studies were evaluated using established critical appraisal tools and individually assessed by two authors to estimate the risk of bias in each study. We were able to synthesize comprehensive descriptive statistics. On the other hand, however, only 65 patients were included, which was considered insufficient to reach a precise conclusion. For a clear association between encephalitis and COVID-19 vaccination, a larger sample size with consistent reporting of clinical phases is required. To reduce the reporting bias, more clinical trials with appropriate follow-up of the treatment protocol must be analyzed. This would give a better insight into the severity and prognosis of the condition. Other limitations inherent to the nature of our systematic review of case reports and series follow naturally and must be considered.

## 5. Conclusions

With the increasing population of vaccinated individuals, a growing body of literature introduced a variety of rare side effects including encephalitis with its various subtypes. Studies included in our report should prompt awareness of possible encephalitis cases with presenting symptoms of abnormal movements, fever, and seizures, particularly 2–3 weeks post-vaccination. Physicians must pay attention to such adverse effects as they can be easily managed if noticed promptly and with excellent recovery rates. Further studies are needed to understand the underlying pathophysiologic mechanism and investigate an association relationship. Due to rarity of reported cases and good overall recovery, we still recommend COVID-19 vaccination.

## Figures and Tables

**Figure 1 vaccines-11-00576-f001:**
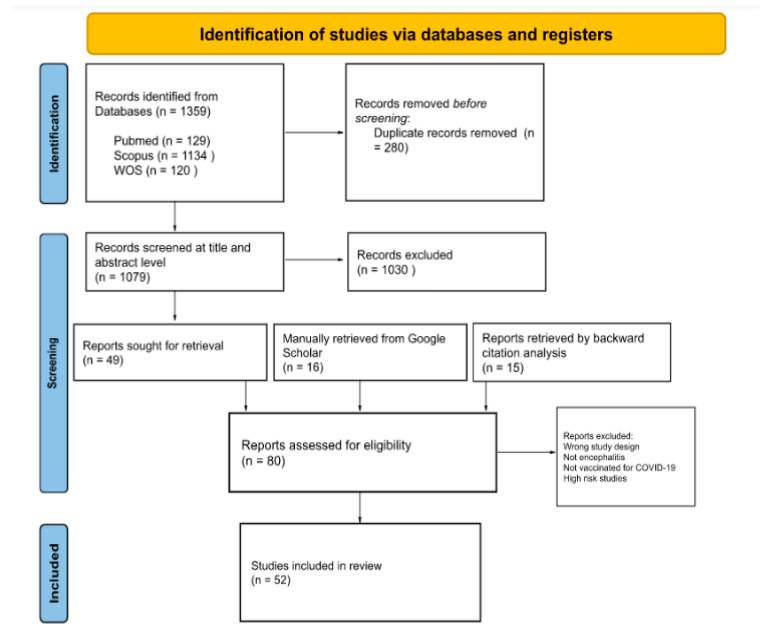
PRISMA flow diagram for included studies.

**Figure 2 vaccines-11-00576-f002:**
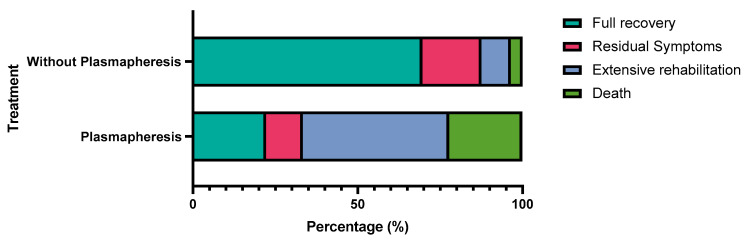
Comparison between the outcome of patients who underwent plasmapheresis versus patients who did not undergo plasmapheresis.

**Table 1 vaccines-11-00576-t001:** Characteristics of included patients.

No.	Author	Vaccine	Age/Sex	Type of Encephalitis	Onset Time (Days)	Dose	Clinical Features	Comorbidities	Other Complaints	Neuroimaging and CSF Analysis	Treatment	Outcome
1	Ahmed et al.	Pfizer BioNTech	61/F	ADEM	7	1	Progressive generalized weakness and difficulty with communication	Hypertension	-	MRI: nonspecific acute versus subacute leukoencephalopathy involving the brainstem and deep white matter	Methylprednisolone, IVIG	Full recovery
2	Ahmed et al.	Pfizer BioNTech	62/F	Meningoencephalitis	5	2	Headache, fever, and rigor for 4 days, inability to stand up and walk, did not obey commands	Ceftriaxone allergy	-	CSF: high protein, pleocytosis	Acyclovir	Full recovery
3	Ahn et al.	AstraZeneca	53/M	AE	7	2	Gait disturbance, dysarthria, cognitive decline, Hoffman sign, and ankle clonus	-	-	MRI: Increased FLAIR signal intensity in the bilateral hippocampus, Multiple enhanced lesions of the spinal cord (C6, T1 level) CSF: pleocytosis, elevated protein, oligoclonal band type 2 (+)	Acyclovir, IVIG, and rituximab	Favorable
4	Albsheer et al.	Moderna	35/F	LE	2	2	Seizures	-	Anisocoria	CSF: pleocytosis CT: temporal lobe hypodensities ANA (+)	Steroids, IVIG, and rituximab	Responded well with no additional neurological sequelae
5	Aljamea et al.	Pfizer BioNTech	38/M	Bickerstaff Brainstem Encephalitis	10	1	Generalized fatigue, weakness, slowing of movement, hypophonia	-	Aspiration pneumonia, constipation	CSF: elevated proteins and albumin, GD1a (+), MRI: T2 hyperintensity within the distal spinal cord/conus medullaris	IVIG, plasmapheresis, and corticosteroids	Management and rehabilitation in a long-term care facility
6	Aljamea et al.	Pfizer BioNTech	54/M	Bickerstaff Brainstem Encephalitis	14	2	Dysphagia, altered mental status, progressive weakness, limb ataxia, disturbed conscious level	DM, hypertension	Aspiration pneumonia, ophthalmoplegia	CSF: pleocytosis, oligoclonal bands (+), GQ1b and GM1 antibodies (+)	IVIG, plasmapheresis, Rituximab	Extensive rehabilitation in a long-term care facility
7	Al-Quliti et al.	AstraZeneca	56/F	ADEM	10		Generalized weakness, lower extremity myalgia, difficulty in the articulation of speech, dysmetria	-	Anorexia	MRI: T2 and FLAIR showed hyperintensities in the subcortical and deep white matter involving basal ganglia CSF: protein and glucose elevated	Hypertonic saline, methylprednisolone	Neck stiffness and bilateral-adduction-gaze deficit were resolved, as well as minimal improvement in her lower limbs’ weakness was observed, able to mobilize freely without assistance
8	Ancau et al.	AstraZeneca	61/M	AHEM	4	1	Fever, headache, apathy, generalized seizure, unconsciousness, bedridden, foaming around the mouth	Hypothyroidism, polymyalgia rheumatica		CT: diffuse hypodense areas in the right subcortical, frontotemporal, and right thalamic region. MRI: bilateral confluent cortical and subcortical FLAIR hyperintense lesions with hemorrhagic involvement of the basal ganglia CSF: moderate disturbance of the BBB	Methylprednisolone, plasmapheresis	Vegetative state
9	Ancau et al.	AstraZeneca	25/F	AHEM	2	1	Cephalgia, fatigue, lack of sensation in legs, paraplegic syndrome, absent tendon reflexes, detrusor areflexia, difficulty urinating, mild weakness, ascending numbness in legs	-	Thoracic back pain	MRI: longitudinal edema along the thoracic spinal cord with contrast enhancement, focal central hemorrhage, bi-hemispheric white matter lesions with focal contrast enhancement CSF: pleocytosis, increased albumin, intrathecal IgM synthesis	Methylprednisolone, Plasmapheresis	Persistent paraplegia
10	Ancau et al.	AstraZeneca	55/F	AHEM	9	1	Meningism, spastic tetraparesis, coma	-	Nausea, dizziness	MRI: multiple FLAIR hyperintense hemorrhagic lesions in the right temporal and parietal lobes, bilaterally in fronto-temporal distribution and in the right occipital lobe and left fronto-basal region CSF: pleocytosis, intrathecal IgM, IgG and IgA, trans-tenetorial herniation, and hydrocephalus occlusion	Right-sided decompressive hemicraniectomy, Methylprednisolone	Death
11	Asaduzzaman et al.	Pfizer BioNTech	15/F	Autoimmune encephalitis	1	2	Fever, agitation, altered consciousness, convulsions	-	Diarrhea, dehydration, palpitation, myocarditis	CSF: pleocytosis, raised protein level, and normal glucose level MRI: no abnormality	Acyclovir, Ceftriaxone, methylprednisolone	Full recovery after 4 weeks
12	Asioli et al.	AstraZeneca	73/F	Anti-LGI1 encephalitis	14	1	FBDS, behavioral disturbances	-	-	CSF: Anti-LGI1 (+), EEG: Bilateral fronto-temporal sharp waves; electrographic temporal seizures, MRI: Bilateral mesial temporal lobe T2-weighted hyper-intensity with swelling in the left hippocampus	Methylprednisolone, Valproate	Seizure-free, normal mental status
13	Asioli et al.	Pfizer BioNTech	66/M	Anti-LGI1 encephalitis	6	2	Cognitive impairment, behavioral disturbances	Hypertension	-	CSF: Anti-LGI1 (+), high protein, RBCs (+), pleocytosis MRI: Bilateral mesial temporal lobe T2-weighted hyper-intensity with swelling and contrast enhancement in the right amygdala and hippocampus, EEG: Right fronto-temporal sharp waves; electrographic temporal seizures	Methylprednisolone, Levetiracetam	Normal mental status in 7 months
14	Asioli et al.	Pfizer BioNTech	66/M	Anti-LGI1 encephalitis	9	2	FBDS, focal seizures, behavioral disturbances	Polyallergy	Hypersomnia	CSF: Anti-LGI1 (+), EEG: Bilateral fronto-temporal epileptiform discharges	Methylprednisolone, Lacosamide	Seizure-free, normal mental status in 3 months
15	Asioli et al.	Moderna	18/F	Anti-LGI1 encephalitis	23	3	Focal seizures, short-term memory impairment	-	-	MRI: Right fronto-temporal sharp waves, CSF: Anti-LGI1 (+)	Methylprednisolone, Lacosamide	Seizure-free, normal mental status in 3 months
16	Autjimanon et al.	Pfizer BioNTech	14/F	Acute encephalopathy	9	1	Fever, headaches, drowsiness, tonic-clonic focal and generalized seizures, status epilepticus	-	-	ANA (+)	Anti-convulsants, immunosuppressants	Discharged home but had residual memory problems
17	Ballout et al.	Pfizer BioNTech	27/M	Autoimmune Encephalitis	6	1	Fatigue, confusion and anxiety, headache, agitation, dysfluent speech with paraphasic errors, and difficulty with writing	-	-	CSF: Pleocytosis with lymphocytic predominance MRI: no abnormalities EEG: mild generalized slowing without epileptiform abnormalities	Methylprednisolone	Full recovery after 1 month
18	Ballout et al.	Moderna	81/M	ADEM	13	1	Change in mental status with severe encephalopathy, fever, absent pupillary response to light, absent right corneal reflex, diffuse hypertonicity, extensor plantar responses bilaterally	-	Fatigue, myalgia	CSF: pleocytosis, elevated protein, elevated myelin basic protein (MBP) MRI: diffusion restriction in the right dorsal medulla with corresponding T2 FlAIR hyperintensity, faint T2 hyperintensities in the left pons, midbrain, and thalamus, and minimal T2 sulcal hyperintensity without contrast enhancement	Methylprednisolone, IVIG therapy, plasmapheresis	Death
19	Bastide et al.	AstraZeneca	49/F	ADEM	2	1	Fever, flu-like symptoms, Lhermitte’s phenomenon, sensory ataxia, Romberg sign, impaired tandem walking, paraparesis, pallesthesia, hypoesthesia, and Sphincter dysfunction	-	-	Brain MRI: large ill-defined T2 FLAIR hyperintensities of periventricular and deep white matter with smaller lesions infratentorially, spared cortex, deep gray matter and subcortical U fibers; SSEPs: abnormal conduction above the sensory decussation in the lower brainstem CSF: mild pleocytosis	Methylprednisolone, rituximab	Improved and MRI showed stability or regression of most lesions
20	Cao et al.	Sinopharm and Sputnik V	24/F	ADEM	14	1	Reduced memory, headache, fever, spasticity, weakness in extremities	-	Loss of appetite	CSF: pleocytosis, oligoclonal band (+) MRI: abnormal signals in the B/L temporal cortex, lesions EEG: epileptiform waves	Immunoglobulin, diazepam, levetiracetam	Full recovery after 1 month
21	Escola et al.	AstraZeneca	43/F	Encephalomyelitis	9	1	Headache, meningism, and fever, sensorimotor tetraparesis, subacute sensorimotor paraparesis, urinary retention, hyperreflexia	Migraine	-	MRI: T2 hyperintense lesions involving frontal cortex, periventricular space, pulvinar thalamic nuclei, brain stem, and cerebellar peduncles CSF: extensive predominant granulocytic pleocytosis, elevated lactate and protein	Methylprednisolone, ceftriaxone, ampicillin, plasma exchange, meropenem, tocilizumab	Patient developed a stuporous to a comatose state and was discharged to a rehabilitation center; after 3 months, she improved with a light cerebellar syndrome (Rt hand intention tremor)
22	Etemadifar et al.	Sinopharm and Sputnik V	50/F	Anti-NMDAR encephalitis	20	2	Behavioral disturbances, muscle pain, limb weakness, ataxia, dizziness, weakness, agitation, (+) Babinski sign	Rituximab-treated MS	Vomiting	MRI: plaques in the periventricular, juxtacortical, and cortical area	Methylprednisolone	Full recovery
23	Fan et al.	Moderna	22/M	AE	6	2	Fever, blurry vision, consciousness disturbance, status epilepticus, slurred speech, memory loss	-	Sinus tachycardia	CSF: no pleocytosis, elevated protein, SARS-CoV-2 spike S1 RBD IgG; EEG: continuous diffuse slowing in theta and delta ranges; CT: mild hypoperfusion in right temporal region	Levetiracetam, acyclovir, valproate sodium, and methylprednisolone	Full recovery
24	Fernandes et al.	Pfizer BioNTech	16/M	Anti-GAD encephalitis	7	1	Generalized tonic clonic seizures	DM	-	EEG: bitemporal focal slowing with admixed sharp waves CSF: pleocytosis, protein elevation	Dexamethasone	Improved with minimal right focal slowing
25	Flannery et al.	Pfizer BioNTech	20–30/F	Anti-NMDAR encephalitis	7	1	Anxiety, decreased mental acuity, insomnia, COVID-19 hypochondria, motor dysfunction, aphasia, accusatory auditory hallucinations, spontaneous defecation, psychosis, catatonia, grand mal seizure, lethargy	Irritable bowels and kidney disease	Tachycardia, hypertension	CSF: mild lymphocyte pleocytosis, anti-NMDA titer 1:20	Olanzapine, haloperidol, lithium, Risperidone, IVIG, methylprednisolone, metoprolol, and rituximab	Improved with minor neurological deficits after 45 days
26	Gao et al.	Moderna	82/F	AE	5	1	Fever, headache, behavioral changes	DM, hypertension	-	EEG: slow waves in the right frontoparietal regions MRI: signal change in the right middle and posterior temporal lobe CSF: elevated protein	Pulse steroid	Full recovery
27	Garibashvili et al.	AstraZeneca	71/M	Anti-LGI1 encephalitis	28	1	Faciobrachial dystonic seizures, mild pallhypaethesia	Heart disease, hypertension and hyperlipidemia	Cardiac pauses	LG1 antibodies: marked increase in serum, normal in CSF	Prednisolone	Full recovery
28	Gogu et al.	Johnson & Johnson	45/M	AHEM	30	1	Right hemiparesis, mixed aphasia, severe headache, agitation, depressed level of consciousness, coma	SARS-CoV-2 (+), DM, Tolosa–Hunt Syndrome	Incomplete left ophthalmoplegia	MRI: multiple ischemic strokes, meningitis, infectious vasculitis, and hemorrhagic encephalitis with extension of the lesion to left fronto-parieto-temporal lobes with hypersignal aspects on the T2, Flair, and DWI images	Methylprednisolone	Death
29	Grossi et al.	Pfizer BioNTech	/M	AE	17	1	Fever, agitation, confusion, headache, and Tonic-Clonic seizures	CLL, herpetic trigeminal rash	Vomiting	CSF: pleocytosis, high albumin, oligoclonal bands (+)	Dexamethasone Acyclovir, Ceftriaxone, Vancomycin, and Levetiracetam	Free from focal neurological impairment after 4 months
30	Huang et al.	AstraZeneca	38/F	Autoimmune Encephalitis	14	1	Acute-onset amnesia, fever, and general malaise for 2 days, incoherent speech, difficulty typing using communication software, tonic–clonic seizure	-	-	MRI: a subacute infarction at the right internal capsule and irregular vascular contour, which indicated a vasculopathy, such as vasculitis CSF: inflammation without pleocytosis EEG: diffuse background slowing with sharp transients at the right temporal region	Levetiracetam + steroid pulse therapy	Full recovery without neurological deficit or sequelae
31	Jarius et al.	Pfizer BioNTech	67/M	MOG encephalomyelitis	10		Color desaturation in the left eye associated with left-sided temporal headache and pain upon eye movement	Arterial hypertension and benign prostate hyperplasia	Optic neuritis	MRI showed swelling and contrast enhancement of the anterior part of the left optic nerves. Visual evoked potentials (VEP) demonstrated prolonged absolute and relative P100 latency and marked amplitude reduction in the left eye (by 40 ms and 73%, respectively, compared with the right eye)	Methylprednisolone	Favorable
32	Kania et al.	Moderna	19/F	ADEM	14	1	Headache, fever, urinary retention	-	Nausea, vomiting, back and neck pain	MRI: multiple hyperintense lesions in T2 weighted and FLAIR images located in both brain hemispheres, pons, the medulla oblongata, and cerebellum CSF: pleocytosis, elevated protein and RBC	Methylprednisolone	Full recovery
33	Kobayashi et al.	Pfizer BioNTech	46/F	Brainstem encehalitis	5	2	Diplopia	Vasculitis	-	MRI: lesion on the dorsal pons across the midline and no gadolinium enhancement	Methyprednisolone	Full recovery
34	Kwon et al.	AstraZeneca	57/F	Autoimmune encephalitis	5	1	Headache, fever, generalized convulsive seizure, cognitive decline including attention and memory deficits along with gradually worsening dysphasia	Hypertension	Myalgia	MRI: restricted diffusion through the left insular and mesial temporal cortices, contrast enhancement CSF: pleocytosis, elevated protein, oligoclonal band (+) EEG: intermittent generalized delta activity	Methylprednisolone, Immunoglobulin, Rituximab	The patient’s language function slowly improved substantially following rituximab therapy, but the memory dysfunction hardly improved. Encephalomalacia change was observed in the left temporal lobe
35	Lazaro et al.	Sinopharm and Sputnik V	26/F	ADEM	28	1	Disorientation, inappropriate behavior, headache, gait imbalance declined memory, hypoprosexia, anosognosia, incoherent speech, visuospatial failures, right upper limb weakness, gait ataxia	-	-	CSF: normal, OCB (+) MRI: nodular hyperintense lesions on T2/FLAIR without restricted diffusion	Methylprednisolone	Full recovery and clear MRI after 3 months
36	Li et al.	AstraZeneca	55/M	AE	7	1	Fever, progressive weakness, consciousness disturbance	Hypertension, hyperlipidemia, and sleep apnea under medication	-	CSF: pleocytosis, elevated protein, ANA (+); high D-dimer	Dexamethasone and subcutaneous fondaparinux	Normal, 4 months later he received Moderna vaccine with no sequelae
37	Maramattom et al.	AstraZeneca	64/M	LE	10	1	Headache, altered sensorium, fever	Glomerulonephritis, Subsegmental pulmonary embolism	-	CT chest; normal MRI brain: hyperintensities in bilateral medial temporal lobe and head and proximal body of hippocampus (L > R) CSF: pleocytosis	Methylprednisolone, plasma exchange, and rituximab	Improved
38	Maramattom et al.	AstraZeneca	46/M	ADEM	5	1	Fever, urinary complaints, progressive paraparesis	A brisk jaw jerk, spastic quadriparesis paresis, loss of posterior column sensations till the T6 level	-	MRI spine: longitudinally extensive transverse myelitis MRI brain: T2, FLAIR hyperintensities in bilateral middle cerebellar peduncle (left > right), pontine tegmentum, right paramedian medulla, and left thalamocapsular region CSF: encephalitis panel: negative	methylprednisolone, plasma exchange	Improved
39	Maramattom et al.	AstraZeneca	64/M	ADEM	20	2	Progressive paresthesia of legs, followed by UL, spastic paraparesis	-	-	MRI brain and spine: bilateral corticospinal tract hyperintensities Dorsal cord hyperintensity at D8–9 CT: normal	methylprednisolone and Rituximab	Improved
40	Maramattom et al.	AstraZeneca	42/F	LE	5	1	Persistent daily headache	-	Papilledema	CSF: opening pressure 32 cm H O2 CSF parameters normal serum and CSF autoimmune MRI: initial MRI: leptomeningeal and sulcal enhancement; 25 days later: large right temporal irregular enhancing lesion with significant perilesional edema	Decompression of lesion Excisional biopsy, prednisolone	Good recovery with some symptoms
41	Monte et al.	Pfizer BioNTech	15/F	Bickerstaff brainstem encephalitis	3	2	Diplopia, dysarthria, consciousness disturbance, fever, asthenia, limb paresthesia, cranial nerve paresis, and gait unsteadiness	Previous recent mycoplasma pneumoniae infection	Cough, vomiting, ophthalmoplegia, abnormal blink reflex	Both normal	Methylprednisolone, immunoglobulins	After 2 weeks, he was able to walk without assistance and the neurological examination was normal. Six weeks after the disease onset, the blink reflex was normal
42	Moslemi et al.	AstraZeneca	27/M	HSE	8	1	Agitation, headache delirium, and disorientation to time, location, and people, slowed psychomotor activity, and loss of alertness	-	Vomiting	CT scans: normal findings with no evidence of involvement MRI: all normal	Acyclovir	Improved
43	Nagaratnam et al.	AstraZeneca	36/F	ADEM	14	1	Headache, fatigue, photophobia, bilateral visual disablement, subjective color desaturation, aching eye movements	-	Photophobia, blurred vision in the right eye	MRI: T1/FLAIR hyperintense lesions involving the subcortical white matter, posterior limb of bilateral internal capsules, pons and left middle cerebellar peduncle, multiple internal punctuate foci of gadolinium contrast enhancement	Methylprednisolone	Improvement in response bilaterally with responses being detectable on the left eye
44	Naz et al.	Sinopharm and Sputnik V	33/F	Anti-NMDAR encephalitis	4	2	Constitutional symptoms, memory disturbance, confusion, fish mouthing movement and seizures	-	-	CSF: pleocytosis, oligoclonal band (+), CT and MRI (−), blood NMDA test (−)	Steroids, IVIG, T-PLEX, and Rituximab	Full recovery
45	Permezel et al.	AstraZeneca	63/M	ADEM	12	1	Vertigo, fatigue, disorientation, declining cognition, impaired attention, poor responsiveness	DM, ischemic heart disease and atrial fibrillation	Abdominal pain, fatigue, ketoacidosis, and silent myocardial infarction	MRI: bilateral foci (>20) of high T2 and FLAIR signal in the white matter	Corticosteroids, plasmapheresis	Death
46	Rastogi et al.	Moderna	59/F	Rhombencephalitis	12	2	Binocular diplopia, paresthesia, hand numbness, decreased sensation, cerebellar dysfunction	Fibromyalgia, migraines, and carpel tunnel syndrome	Dizziness, lethargy	CSF: elevated protein, elevated glucose, lymphocytic pleocytosis; MRI: multiple focal poorly defined regions of contrast enhancement in the cerebral cortex, deep grey matter, brainstem, and cerebellum	Expectant, without empiric corticosteroids or antimicrobials	Ongoing gradual improvement
47	Rinaldi et al.	AstraZeneca	45/M	ADEM	12	1	Numbness of limbs, trunk and legs, slurred speech, difficulty swallowing, clumsy right-hand movements, dysarthria, dysphagia, urge incontinence	-	Reduced visual acuity	MRI: large, poorly marginated T2-weighted hyperintensities in the pons, right cerebellar peduncle, right thalamus, and multiple spinal cord segments. All lesions, except the thalamic one and a single dorsal spinal area, showed blurred gadolinium enhancement on T1-weighted images CSF: pleocytosis	Methylprednisolone	Complete clinical recovery and no relapses, almost entire resolution of the brainstem and spinal cord lesions at the dorsal/conus medullaris level, and further shrinkage of cervical areas
48	Saad et al.	Pfizer BioNTech	69/F	Acute encephalopathy	5	1	Coma, seizure, status epilepticus	-	-	CSF: high protein, MRI: pyriform-pattern diffusion restriction in the right hemisphere and left frontoparietal region	Methylprednisolone, antibiotics, and antivirals	Discharged in a deeply comatose status on day 30 of hospital admission
49	Sawczyńska et al.	Unknown	77/F	AE	14	1	Fever, attention and cognition disturbances, confusion, hyperactivity, delusions, chorea, orofacial dyskinesia, psychomotor slowing, seizures, hemiparesis, loss of consciousness	COVID-19 infection, hypertension, DM, hypothyroidis, urinary incontinence, and multiple malignancies in remission, slight cognition disturbances	Atrial fibrillation, pneumonia, sepsis, pulmonary embolism, urinary tract infection, reactive arthritis	MRI: features of cerebral small vessel disease, diffuse white matter hyperintensities, cortical and subcortical atrophy EEG: FIRDA pattern	Methylprednisolone, diazepam, remdesivir, IVIG, and antiepileptic	Hospitalization for non-neurological complications
50	Senda et al.	Pfizer BioNTech	72/F	Acute Meningoencephalitis	3	1	Depressed level of consciousness, headache	Rheumatoid vasculitis, DM, and hyperlipidemia	General fatigue	CSF: a cell count of four cells/mm^3^ (all mononuclear leukocytes), an increased protein level, IgG index was elevated (1.13) MRI: hyperintensities in white matter of the bilateral frontotemporal areas on DWI, more on the right side FLAIR images: diffuse cerebral cortex swelling in bilateral frontotemporal areas, also stronger on the right side	Intravenous steroid pulse and gammaglobulin therapies	Improved
51	Shinet al.	AstraZeneca	35/F	Autoimmune encephalitis	5	1	Dysarthria, abnormal movements, anxiety, fever, rigidity, dystonia, catatonia, motor aphasia, jaw-opening dystonia, hypophonia, drooling, reduced voluntary movements	-	Sinus tachycardia	MRI: swelling of the hippocampus, encephalomalacia in frontoparietal lobes EEG: diffuse beta wave activity, intermittent generalized delta activity	Methylprednisolone, immunoglobulins, rituximab	After 1 week, her catatonia, rigidity, and drooling had improved, and she could walk for a short distance without assistance; however, she still had significant rigidity
52	Shyu et al.	Moderna	58/F	AE	7	1	Fever, cognitive deficits, left deviation of the head and eyeballs, and mild weakness of the right UL	-	-	CSF: pleocytosis, elevated protein, CSF/serum albumin ratio of 19.7	Dexamethasone	Regained normal cognitive function and was discharged in 13 days
53	Shyu et al.	Moderna	21/M	AE	7	1	Coma, status epilepticus	-	-	CSF: elevated protein and microalbumin EEG: continuous diffuse slowing in the theta and delta ranges, indicating moderate diffuse cerebral dysfunction SPECT: hypoperfusion in the right temporal region	Methylprednisolone	Healthy and seizure free after 3 months
54	Sluyts et al.	Moderna	48/3	AE	6	booster dose	Agitation, physical aggression, mutism, left arm: paretic and atactic, confusion	-	Bradyphrenia	CSF: pleocytosis, elevated protein MRI: small left internal capsule developmental venous anomaly	Ceftriaxone, amoxicilline, and acyclovir	Full recovery after 3 days from steroids admission
55	Takata et al.	AstraZeneca	22/F	Autoimmune encephalitis	Few	2	Headache, fatigue, confusion, agitation, hallucinations, fever, disorientation	-	-	CSF: opening pressure of 30 cm H_2_O, pleocytosis, IgG oligoclonal bands (+ve)	Ceftriaxone, acyclovir, lorazepam, and olanzepine	She remains on low-dose olanzapine and is functionally well with independent activities of daily living, but her family reports that she has not recovered back to her pre-morbid state
56	Torrealb a-Acosta G et al.	Moderna	77/M	Meningoencephalitis	2	1	Dizziness, fever, rashes, headache, double vision, confusion	Coronary artery disease, hyperlipidemia, and hypothyroidism	Edematous erythematous papules and plaques with overlying pustules on the trunk and abdomen	CSF: pleocytosis, increased protein vEEG: generalized slow theta range with state changes and reactivity	Methylprednisolone following prednisone	Full recovery
57	Vences et al.	Pfizer BioNTech	72/M	AE	1	1, relapse in 4 2	Malaise, headache, fever, confusion, aggressiveness, and gait alterations	-	-	CSF: elevated protein MRI: circumscribed encephalitis at the anterior frontal and bilateral temporal lobes	Methylprednisolone	Favorable
58	Vogrig et al.	Pfizer BioNTech	56/F	ADEM	14	1	Malaise, chills, unsteadiness of gait	-	-	MRI: hyperintensities on FLAIR sequences involving the left cerebellar peduncle, with moderate mass effect on the fourth ventricle	Prednisone	Full recovery
59	Walter et al.	Pfizer BioNTech	30/M	RE	21	2	Malaise, headache, taste disorder, facial paralysis (left side), gait disturbance by ataxia, hypoglossal nerve paralysis	-	-	MRI: weak FLAIR hyperintensity of the brainstem, mesencephalon and cerebellar around the fourth ventricle without contrast enhancement CSF: pleocytosis	Methylprednisolone	Full recovery within a few weeks
60	Werner et al.	Pfizer BioNTech	35/F	Autoimmune encephalitis	2	2	Fever, headache, visual impairment, behavioral changes, recurrent focal to bilateral tonic-clonic seizures, reduced level of consciousness, and choreatic movements	-	Skin rash	Cerebral magnetic resonance imaging: swelling in the (para-) hippocampal region predominantly on the left hemisphere and bilateral subcortical subinsular FLAIR hyperintensities. Cerebrospinal fluid analysis: a lymphocytic pleocytosis of 7 cells/μL and normal protein and immunoglobulin parameters	Levetiracetam, lacosamide, methylprednisolone, and plasma exchange	Partial recovery
61	Yazdanpanah et al.	Sinopharm and Sputnik V	37/M	ADEM	Few	1	Intermittent myalgia, drooling, progressive weakness of 4 limbs, bilateral f, dysphagia	-	Nausea, vomiting	Brain MRI: Hyperintense foci within the left cerebral peduncle, left corticospinal tract, right and left sides of pons and medulla Spine MRI: unremarkable Magnetic resonance spectroscopy (MRS): confirmed the demyelination process by the presence of Myoinositol and Choline peaks	Heparin, Pantoprazole, Clindamycin, Paracetamol, and Methylprednisolone	Full recovery
62	Zlotnik et al.	Pfizer BioNTech	48/M	Anti-LGI1 encephalitis	18	2	Fatigue, memory deficit, anterograde amnesia	-	-	MRI: hyperintense signal on both medial temporal lobes	Methylprednisolone	Recovered, but still faces some executive skills difficulties
63	Zuhorn et al.	AstraZeneca	21/F	Autoimmune encephalitis	5	1	Headache, concentration difficulties, fever, malaise, epileptic seizure	-	-	CSF: pleocytosis EEG: diffuse slow theta rhythm	Dexamethasone	Normal state of the parenchyma without sequelae
64	Zuhorn et al.	AstraZeneca	63/F	Autoimmune encephalitis	6	-	Gait deterioration, twitching, opsoclonus-myoclonus syndrome	‘	Oral anticoagulation, vigilance disorder	EEG: diffuse slow theta rhythm CSF: pleocytosis	Methylprednisolone	Normal state of the parenchyma
65	Zuhorn et al.	AstraZeneca	63/M	Autoimmune encephalitis	8	-	Fever, aphasia	-	-	CSF: pleocytosis	-	Further improvement could be observed, no evidence of structural lesions

Normal IgG index < 0.66. UL: upper limb, FBDS: Faciobrachial dystonic seizures, DM: diabetes mellitus, PD: Parkinson disease, MS: Multiple sclerosis.

**Table 2 vaccines-11-00576-t002:** Summary of patients’ characteristics, common symptoms, laboratory and imaging findings, treatment, and treatment outcomes of patients with post-COVID-19 vaccine encephalitis.

Variable		Descriptive Statistics
Sex (%)	Male	36 (55.4%)
	Female	28 (43.1%)
	Transgender male	1 (1.5%)
Age (mean ± SD)		46.82 ± 19.25
Period after vaccination in days (mean ± SD)		9.97 ± 7.16
Vaccine Subtypes (%)	AstraZeneca	25 (38.5%)
	Pfizer BioNTech	22 (33.8%)
	Moderna	11 (16.9%)
	Sinopharm and Sputnik	5 (7.7%)
	Johnson & Johnson	1 (1.5%)
	Unknown	1 (1.5%)
Vaccine dose (%)	1st	41 (66.1%)
	2nd	18 (29%)
	3rd	1 (1.6%)
	4th	2 (3.2%)
Encephalitis Subtypes (%)	Acute encephalitis	11 (16.9%)
	ADEM	14 (21.5%)
	AHEM	4 (6.2%)
	Other	36 (55.4%)
Headache (%)		20 (30.8 %)
Fever (%)		23 (35.4 %)
Seizure (%)		15 (23.1 %)
Abnormal movement (%)		24 (36.9 %)
CSF findings (%)		
	Pleocytosis	32 (49.2 %)
	High protein	7 (10.8 %)
	Antibodies positive	6 (9.2 %)
MRI findings (%)		
	Abnormal	40 (61.5 %)
Treatment (%)	Steroids	56 (86.2 %)
	Immunoglobulins	15 (23.1 %)
	Plasmapheresis	9 (13.8 %)
	Antiviral	10 (15.4 %)
	Immunosuppressive drug	53 (81.5 %)
Treatment outcome (%)	Full recovery	41 (63.1 %)
	Residual Symptoms	11 (16.9 %)
	Extensive rehabilitation	9 (13.8 %)
	Death	4 (6.2 %)
	Death-associated comorbidities	Hemicraniectomy, Tolosa Hunt Syndrome, Diabetes type 2, ischemic heart disease, atrial fibrillation

**Table 3 vaccines-11-00576-t003:** Relation between the type of vaccine and the outcome of treatment.

	Outcome					
Vaccine	Full Recovery	Residual Symptoms	Extensive Rehabilitation	Death	Total	*p*-Value
AstraZeneca	13	4	6	2	25	0.124 *
	52.0%	16.0%	24.0%	8.0%	100.0%	
Pfizer BioNTech	14	5	3	0	22	
	63.6%	22.7%	13.6%	0.0%	100.0%	
Moderna	9	1	0	1	11	
	81.8%	9.1%	0.0%	9.1%	100.0%	
Sinopharm and Sputnik V	5	0	0	0	5	
	100.0%	0.0%	0.0%	0.0%	100.0%	
Johnson & Johnson	0	0	0	1	1	
	0.0%	0.0%	0.0%	100.0%	100.0%	
Unknown mRNA vaccine	0	1	0	0	1	
	0.0%	100.0%	0.0%	0.0%	100.0%	

* Fischer’s exact test.

**Table 4 vaccines-11-00576-t004:** Relation between the subtypes of encephalitis and the outcome of treatment.

	Outcome					
Encephalitis Subtype	Full Recovery	Residual Symptoms	Extensive Rehabilitation	Death	Total	*p*-Value
Acute encephalitis	10	1	0	0	11	<0.001 *
	90.9%	9.1%	0.0%	0.0%	100.0%	
ADEM	10	2	1	2	15	
	66.7%	13.3%	6.7%	13.3%	100.0%	
AHEM	0	0	2	2	4	
	0.0%	0.0%	50.0%	50.0%	100.0%	
Other	21	8	6	0	35	
	60.0%	22.9%	17.1%	0.0%	100.0%	

* Chi-square test.

**Table 5 vaccines-11-00576-t005:** Comparison between the outcome of patients who underwent plasmapheresis versus patients who did not undergo plasmapheresis.

	Outcome					
Plasmapheresis	Full Recovery	Residual Symptoms	Extensive Rehabilitation	Death	Total	*p*-Value
Plasmapheresis	2	1	4	2	9	0.002 *
	22.2%	11.1%	44.4%	22.2%	100.0%	
Without Plasmapheresis	39	10	5	2	56	
	69.6%	17.9%	8.9%	3.6%	100.0%	

* Chi-square test.

**Table 6 vaccines-11-00576-t006:** Joanna Brigg’s institute critical appraisal checklist for case reports.

Author’s Name	Q1	Q2	Q3	Q4	Q5	Q6	Q7	Q8	Total	%	ROB
Ahmed et al.	No	Yes	Yes	Yes	Yes	Yes	No	Yes	6	75%	Low risk
Ahmed et al.	No	No	Yes	Yes	Yes	Yes	No	Yes	5	63%	Moderate risk
Ahn et al.	No	No	Yes	Yes	Yes	Yes	Yes	Yes	6	75%	Low risk
Albsheer et al.	No	Yes	Yes	Yes	No	No	No	Yes	4	50%	Moderate risk
Aljamea et al.	UC	Yes	Yes	Yes	Yes	Yes	Yes	Yes	7	88%	Low risk
Al-Quliti et.al	No	Yes	Yes	Yes	Yes	No	No	Yes	5	63%	Moderate risk
Ancau et al.	No	Yes	Yes	Yes	Yes	Yes	Yes	Yes	7	88%	Low risk
Asaduzzaman et al.	No	Yes	Yes	Yes	Yes	Yes	Yes	Yes	7	88%	Low risk
Autjimanon et al.	No	Yes	Yes	Yes	Yes	Yes	UC	Yes	6	75%	Low risk
Bastide et al.	No	Yes	Yes	Yes	Yes	Yes	Yes	Yes	7	88%	Low risk
Cao et al.	No	Yes	Yes	Yes	Yes	Yes	UC	Yes	6	75%	Low risk
Ebadi et al.	No	Yes	Yes	No	No	No	No	Yes	3	38%	High risk
Escolà et al.	No	Yes	Yes	Yes	Yes	Yes	Yes	Yes	7	88%	Low risk
Etemadifaret et al.	No	Yes	Yes	Yes	Yes	Yes	Yes	Yes	7	88%	Low risk
Fan et al.	No	Yes	Yes	Yes	Yes	Yes	No	Yes	6	75%	Low risk
Fernandes et al.	No	Yes	Yes	Yes	Yes	Yes	Yes	Yes	7	88%	Low risk
Flannery et al.	No	Yes	Yes	Yes	Yes	Yes	No	Yes	6	75%	Low risk
Gao et al.	No	Yes	Yes	Yes	Yes	Yes	No	Yes	6	75%	Low risk
Garibashvili et al.	No	Yes	Yes	Yes	Yes	Yes	No	Yes	6	75%	Low risk
Gogu et al.	Yes	Yes	Yes	Yes	Yes	Yes	Yes	Yes	8	100%	Low risk
Grossi et al.	Yes	Yes	Yes	Yes	Yes	Yes	Yes	Yes	8	100%	Low risk
Huang et al.	No	Yes	Yes	Yes	Yes	Yes	No	Yes	6	75%	Low risk
Jarius et al.	Yes	Yes	Yes	Yes	Yes	Yes	No	Yes	7	88%	Low risk
K. Kania et al.	No	Yes	Yes	Yes	No	Yes	UC	Yes	5	63%	Moderate risk
Kobayashi et al.	Yes	Yes	Yes	Yes	Yes	Yes	No	Yes	7	88%	Low risk
Kobayashi et al.	No	Yes	Yes	Yes	Yes	Yes	No	Yes	6	75%	Low risk
Kwon et al.	Yes	Yes	Yes	Yes	Yes	Yes	Yes	Yes	8	100%	Low risk
Lagioia et al.	No	Yes	No	No	No	No	No	No	1	13%	High risk
Lazaro et al.	No	Yes	Yes	Yes	Yes	Yes	No	Yes	6	75%	Low risk
Li et al.	No	Yes	Yes	Yes	Yes	Yes	No	Yes	6	75%	Low risk
Mörz et al.	No	Yes	Yes	No	No	No	No	Yes	3	38%	High risk
Moslemi et al.	No	Yes	Yes	Yes	Yes	Yes	Yes	Yes	7	88%	Low risk
Nagaratnam et al.	No	Yes	Yes	Yes	Yes	Yes	Yes	Yes	7	88%	Low risk
Naz et al.	No	Yes	Yes	Yes	Yes	Yes	UC	Yes	6	75%	Low risk
Permezel et al.	No	Yes	Yes	Yes	Yes	Yes	Yes	Yes	7	88%	Low risk
Rastogi et al.	No	Yes	Yes	Yes	Yes	Yes	UC	Yes	6	75%	Low risk
Rinaldi et al.	No	Yes	Yes	Yes	Yes	Yes	No	Yes	6	75%	Low risk
Saad et al.	No	No	Yes	No	Yes	Yes	Yes	Yes	5	63%	Moderate risk
Sawczyńska et al.	No	No	Yes	Yes	Yes	Yes	Yes	Yes	6	75%	Low risk
Senda et al.	Yes	Yes	Yes	Yes	Yes	Yes	No	Yes	8	100%	Low risk
Shin HR et al.	No	Yes	Yes	Yes	Yes	Yes	Yes	Yes	8	100%	Low risk
Shyu et al.	No	No	Yes	Yes	Yes	Yes	No	Yes	5	63%	Moderate risk
Sluyts et al.	No	Yes	Yes	Yes	No	No	No	Yes	4	50%	Moderate risk
Takata et al.	Yes	Yes	Yes	Yes	Yes	Yes	Yes	Yes	8	100%	Low risk
Torrealba-Acosta et al.	No	Yes	Yes	Yes	No	Yes	UC	Yes	5	63%	Moderate risk
Vences et al.	No	Yes	Yes	Yes	Yes	Yes	Yes	Yes	7	88%	Low risk
Vogrig et al.	No	Yes	Yes	Yes	Yes	Yes	Yes	Yes	7	88%	Low risk
Walter et al.	No	No	Yes	Yes	Yes	Yes	NA	Yes	5	63%	Moderate risk
Werner et al.	No	Yes	Yes	Yes	Yes	Yes	UC	Yes	6	75%	Low risk
Yazdanpanah et al.	Yes	Yes	Yes	Yes	No	Yes	No	Yes	6	75%	Low risk
Zlotnik et al.	No	Yes	Yes	Yes	Yes	Yes	Yes	Yes	7	88%	Low risk
Zuhorn et al.	No	Yes	Yes	Yes	Yes	Yes	No	Yes	6	75%	Low risk

**Table 7 vaccines-11-00576-t007:** Joanna Brigg’s institute critical appraisal checklist for case series.

Author’s Name	Q1	Q2	Q3	Q4	Q5	Q6	Q7	Q8	Q9	Q10	Total	%	ROB
Asioli et al.	Yes	Yes	Yes	Yes	Yes	No	UC	Yes	Yes	NA	7	70%	Moderate risk
Ballout et al.	Yes	No	Yes	Yes	Yes	No	Yes	No	Yes	NA	6	60%	Moderate risk
Maramattom et al.	Yes	UC	Yes	No	UC	UC	No	Yes	Yes	Yes	5	50%	Moderate risk

## Data Availability

The data that support the findings of this systematic review are available from the original studies, but restrictions may apply. Some authors may not have provided open access to their data. All data are available upon reasonable request and with permission of the original authors.

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
