# Peer review of "Encephalitis following COVID-19 Vaccination: A Systematic Review"

_vaccines, 2023, doi:10.3390/vaccines11030576_

Round 1

Reviewer 1 Report

The topic of side effects with the new anti-SARS-COV-2-vaccination strategies (mRNA and Adenovector based vaccines) is of general importance, however, I think the authors, in this manuscript, are trying to walk a path between “political correctness” and science. Therefore, even though the study is important it needs purging from any “political correctness” and the scientific facts have to be brought forward in the abstract, introduction and discussion.

It is very disturbing that the authors were able to identify and compile 52 different peer-reviewed publications from 65 cases of the new expression vaccines (mRNA and Adenovector based) associated with the severe disease of encephalitis. How many unreported such cases are there? How long after the vaccination an encephalitis case was ascribed to side effects of the vaccination? It seems not more than 30 days from this study.

The authors tried to answer in the discussion with citing reference 24 the occurrence of encephalitis side effect with the new vaccination strategies. Nevertheless, these estimates need a revisit. One has to take into account two major points of critic: 1) occurrence incidences of encephalitis vaccination side effects are based on delivered vaccination doses and not on case numbers. In general scientists knew only on how many doses vaccine were delivered  from the companies per country and not how many were really injected. 2) These estimates are further based on about five months vaccination period at the start of the vaccination campaign where almost no vaccination side effects were reported as the vaccination was seen as side effect free. These estimates are therefore useless.

I would therefore focus more on the general occurrence of special events in correlation to the time of the mass vaccination program rollout e.g. the data analyst T. Lausen in Germany about the health insurance KBV codes, which show that about 2.5 million had to be treated either by a medical doctor or in the hospitals of about 75 Mio people.

Or about the excess deaths in the OECD. Stat data base.

Or about the Lincoln National Life Insurance Company: they are in the Life insurance busyness in the USA and they had to pay out in 2019 about 500 mio Dollars, in 2020 about 548mio Dollars, this was with beginning of the pandemic and the more pathogenic variants, and in 2021, with the mass rollout of the vaccination program, 1.4 billion dollars had to be payed out. This is an increase of life insurance pay outs of 163% from 2020 to 2021 and this correlates with the mass vaccination program in the US.

Even in the original Pfizer vaccination trial one identifies scientific irregularities: statistical tricks and time limits artificially were introduced to pump up vaccination effectivity and vaccination side effect monitoring was also cut short in vaccination of the placebo group.

Red flags should occur if one reads about the risk/ benefit ratio of mRNA vaccines in light of the most recent publication in Vaccine (Serious adverse events of special interest following mRNA COVID-19 vaccination in randomized trials in adults; Joseph Fraiman et al., 2022). This publication is the logical outcome following previous publications that indicate that the expression concentration of the spike protein is uncontrollable (Röltgen et al., 2022, Cell; https://www.cell.com/cell/fulltext/S0092-8674(22)00076-9?rss=yes) and that mRNA of the vaccine is found in the germinal centers of the secondary lymph organs (Röltgen et al., 2022, Cell), possibly disturbing the adaptive immune responses generally (Lederer et al., 2022, Cellhttps://pubmed.ncbi.nlm.nih.gov/35202565/).

Or about the relative vaccination effectivity reported: COVID-19 vaccine efficacy and effectiveness in Lancet Microbe (https://pubmed.ncbi.nlm.nih.gov/33899038/)

Or recent study indicating that with every additional vaccine dose the risk increases to contract COVID-19 (https://www.medrxiv.org/content/10.1101/2022.12.17.22283625v1.full.pdf).

Consequently these phrases in quotations need to be rewritten or removed:

Due to good overall recovery, we still recommend COVID-19 vaccination“. As you indicated 37% did not fully recover and this you call a good recovery?

These vaccines are effective through four various mechanisms that share the aim of eliciting immune response:

Recently, COVID-19 emerged as a new public health crisis affecting worldwide populations. Since the outbreak in late 2019, 627 million confirmed cases and 6.5 million deaths have been reported3.“ Reference 3 in the manuscript is listed as “Unable to find information for 13918430.

These numbers about the cases and deaths caused by SARS-COV-2 are from the WHO and are not scientifically sound. Why do I say this? There are different evidences that reveal major problems with these numbers. Briefly, these numbers are chiefly based on the RT-PCR assay testing. However, CT values were not used in the categorization of virus positivity or infectivity. Anyway, infectivity cannot be determined by these assays. Subsequently, the case and death numbers were based purely on a yes or no positivity of the mucosal assays. The mucosal assays were also not evaluated carefully through a ROC analysis to investigate the usefulness of the assays in the population analysis. As it is also known that coronaviruses are often seen with other major pathogens of the respiratory tract these cases or deaths were not stringently analyzed for other coinfections  e.g. adenovirus, H1N1, Streptococcus pneumonia, chlamydia pneumonia and many others.

And even the simplest scientific due diligent to identify deaths were neglected to identify a person whether she/he had died on SARS-COV-2 or with the virus only.

These WHO numbers I would not even bring indirectly into science: they are at best useless.

Author Response

Reviewer 1 report

Comment: The topic of side effects with the new anti-SARS-COV-2-vaccination strategies (mRNA and Adenovector based vaccines) is of general importance, however, I think the authors, in this manuscript, are trying to walk a path between "political correctness" and science. Therefore, even though the study is important it needs purging from any "political correctness" and the scientific facts have to be brought forward in the abstract, introduction and discussion.

Response: Thank you for your precious feedback on the manuscript. We reported data published under a rigorous peer-review in indexed journals and strict publisher guidelines. It is true that in a scientific context, only objective scientific results should be reported, however, amidst the current COVID-19 pandemic, concerns are raised about the misuse of published COVID-19 vaccine data for antivaccine campaigns and the context of "evidence" that can go against the CDC and WHO recommendations for critical COVID-19 cases.

Comment: It is very disturbing that the authors were able to identify and compile 52 different peer-reviewed publications from 65 cases of the new expression vaccines (mRNA and Adenovector based) associated with the severe disease of encephalitis. How many unreported such cases are there? How long after the vaccination an encephalitis case was ascribed to side effects of the vaccination? It seems not more than 30 days from this study.

Response: We understand and regret the disturbance our methodology might have caused. We searched main electronic databases reporting cases from all over the globe through a thorough screening procedure incorporating six authors in three independent screening teams. Any systematic review has an inherent risk of a reporting bias. However, we supplemented our search with a manual search for published articles in grey literature, performed backward citation analysis, and ranked these extracted studies using thorough quality assessment before inclusion. Understandably, post-vaccination encephalitis occurs within the first few weeks when immunologic reaction against the vaccine occurs. This is proposed as one of the theories for underlying pathology discussed in the paper. Moreover, to the extent of our knowledge, no reported encephalitis cases in the literature reported side effect onset of more than 1 month.

Comment: The authors tried to answer in the discussion with citing reference 24 the occurrence of encephalitis side effect with the new vaccination strategies. Nevertheless, these estimates need a revisit. One has to take into account two major points of criticism: 1) occurrence incidences of encephalitis vaccination side effects are based on delivered vaccination doses and not on case numbers. In general, scientists knew only on how many doses of vaccine were delivered from the companies per country and not how many were really injected. 2) These estimates are further based on about five months vaccination period at the start of the vaccination campaign where almost no vaccination side effects were reported as the vaccination was seen as side effect free. These estimates are therefore useless.

Response: Thank you for your notice. We considered an edit to this important comment.

Comment: I would therefore focus more on the general occurrence of special events in correlation to the time of the mass vaccination program rollout e.g. the data analyst T. Lausen in Germany about the health insurance KBV codes, which show that about 2.5 million had to be treated either by a medical doctor or in the hospitals of about 75 Mio people.

Or about the excess deaths in the OECD. Stat data base.

Or about the Lincoln National Life Insurance Company: they are in the Life insurance busyness in the USA and they had to pay out in 2019 about 500 mio Dollars, in 2020 about 548mio Dollars, this was with beginning of the pandemic and the more pathogenic variants, and in 2021, with the mass rollout of the vaccination program, 1.4 billion dollars had to be payed out. This is an increase of life insurance pay outs of 163% from 2020 to 2021 and this correlates with the mass vaccination program in the US.

Even in the original Pfizer vaccination trial one identifies scientific irregularities: statistical tricks and time limits artificially were introduced to pump up vaccination effectivity and vaccination side effect monitoring was also cut short in vaccination of the placebo group.

Red flags should occur if one reads about the risk/ benefit ratio of mRNA vaccines in light of the most recent publication in Vaccine (Serious adverse events of special interest following mRNA COVID-19 vaccination in randomized trials in adults; Joseph Fraiman et al., 2022). This publication is the logical outcome following previous publications that indicate that the expression concentration of the spike protein is uncontrollable (Röltgen et al., 2022, Cell; https://www.cell.com/cell/fulltext/S0092-8674(22)00076-9?rss=yes) and that mRNA of the vaccine is found in the germinal centers of the secondary lymph organs (Röltgen et al., 2022, Cell), possibly disturbing the adaptive immune responses generally (Lederer et al., 2022, Cell; https://pubmed.ncbi.nlm.nih.gov/35202565/).

Or about the relative vaccination effectivity reported: COVID-19 vaccine efficacy and effectiveness in Lancet Microbe (https://pubmed.ncbi.nlm.nih.gov/33899038/)

Or recent study indicating that with every additional vaccine dose, the risk increases to contract COVID-19 (https://www.medrxiv.org/content/10.1101/2022.12.17.22283625v1.full.pdf).

Consequently, these phrases in quotations need to be rewritten or removed:

"Due to good overall recovery, we still recommend COVID-19 vaccination" As you indicated 37% did not fully recover and this you call a good recovery?

"These vaccines are effective through four various mechanisms that share the aim of eliciting immune response."

Response: Thank you for your suggestion. We considered and edited these sentences accordingly.

Comment: "Recently, COVID-19 emerged as a new public health crisis affecting worldwide populations. Since the outbreak in late 2019, 627 million confirmed cases and 6.5 million deaths have been reported3". "Reference 3 in the manuscript is listed as "Unable to find information for 13918430".

Response: Thanks for the notice, reference was corrected.

Comment: These numbers about the cases and deaths caused by SARS-COV-2 are from the WHO and are not scientifically sound. Why do I say this? There are different evidences that reveal major problems with these numbers. Briefly, these numbers are chiefly based on the RT-PCR assay testing. However, CT values were not used in the categorization of virus positivity or infectivity. Anyway, infectivity cannot be determined by these assays. Subsequently, the case and death numbers were based purely on a yes or no positivity of the mucosal assays. The mucosal assays were also not evaluated carefully through a ROC analysis to investigate the usefulness of the assays in the population analysis. As it is also known that coronaviruses are often seen with other major pathogens of the respiratory tract these cases or deaths were not stringently analyzed for other coinfections e.g. adenovirus, H1N1, Streptococcus pneumonia, chlamydia pneumonia and many others.

And even the simplest scientific due diligence to identify deaths were neglected to identify a person whether she/he had died on SARS-COV-2 or with the virus only. These WHO numbers I would not even bring indirectly into science: they are at best useless.

Response: Thank you very much for the great insights. We will definitely consider it.

Reviewer 2 Report

This is an interesting review that comprises the topic of encephalitis induced by COVI-19 vaccination. This topic is of great interest for the scientific community since the vaccination now is extensively administered and several cases of encephalitis derived by COVID-19 vaccination could be under diagnosed. The review contents papers since the onset of the pandemia to October 2022. The tables and figure are appropriate, and the text is easy to follow.  

Author Response

Thank you so much for your time spent on this great peer review